# Long-Term Persistence of Mitochondrial DNA Instability among HCV-Cured People Who Inject Drugs

**DOI:** 10.3390/biomedicines10102541

**Published:** 2022-10-12

**Authors:** Mélusine Durand, Nicolas Nagot, Quynh Bach Thi Nhu, Amélie Vizeneux, Linh Le Thi Thuy, Huong Thi Duong, Binh Nguyen Thanh, Delphine Rapoud, Roselyne Vallo, Catherine Quillet, Hong Thi Tran, Laurent Michel, Thanh Nham Thi Tuyet, Oanh Khuat Thi Hai, Vinh Vu Hai, Jonathan Feelemyer, Philippe Vande Perre, Don Des Jarlais, Khue Pham Minh, Didier Laureillard, Jean-Pierre Molès

**Affiliations:** 1Pathogenesis and Control of Chronic and Emerging Infections, University of Montpellier, INSERM, 34000 Montpellier, France; 2Faculty of Public Health, Hai Phong University of Medicine and Pharmacy, Hai Phong 180000, Vietnam; 3Pierre Nicole Center, French Red Cross, 75005 Paris, France; 4Supporting Community Development Initiatives, Hanoi 111000, Vietnam; 5Infectious Diseases Department, Viet Tiep Hospital, Hai Phong 180000, Vietnam; 6College of Global Public Health, New York University, New York, NY 10012, USA; 7Infectious Diseases Department, Caremeau University Hospital, 30029 Nîmes, France

**Keywords:** HCV, DAA, mitochondria, genotoxicity, side effect

## Abstract

People who inject drugs (PWID) are a population exposed to many genotoxicants and with a high prevalence of HCV infection. Direct-acting antiviral (DAA) regimens are now widely used to treat chronic HCV infection. Although side effects to treatment are currently rare, the long-term effects such as suspicions of de novo hepatocellular carcinoma (HCC) occurrence or HCC recurrence and cardiac defects are still up for debate. Given the structure of DAAs, the molecules have a potential mitochondrial DNA (mtDNA) genotoxicity. We have previously reported acute mtDNA toxicity of three DAA regimens among PWID with a strong impact on the rate of mtDNA deletion, less on the quantity of mtDNA copy per cell at sustained viral response at 12 weeks (SVR12). Herein, we report the mtDNA parameters nine months after drug discontinuation. We observed that the percentage of the deleted mtDNA genome increased over time. No exposure to any other genotoxicants during this period was associated with a high deletion percentage, suggesting that the replicative advantage of the deleted molecules outweighed their elimination processes. Such observation calls for longer-term follow-up and may contribute to the molecular basis of subclinical side effects of DAA treatments.

## 1. Introduction

The availability of direct-acting antivirals (DAA) has revolutionized HCV therapy. Even though we lack sufficient hindsight on their long-term side effects, there are still controversial data on adverse cardiac side effects, higher extrahepatic and intrahepatic malignancies, accelerated hepatocellular carcinoma (HCC) recurrence and aggressiveness, as well as doubts of genotoxicity [1,2,3]. The molecular basis of these side effects has not been elucidated, however, DAAs that are nucleoside analogues could act as substrates for human polymerases, including mitochondrial RNA and DNA polymerases [4,5]. Unlike nuclear polymerases, mitochondrial DNA (mtDNA) polymerases are more sensitive to genotoxicants due to their lack of editing functions. Recent in vitro models showed that sofosbuvir (SOF) and daclatasvir (DCV) drugs impaired mitochondrial morphologies and lowered the mtDNA copy number in yeast [3]. Furthermore, the association of sofosbuvir and ribavirin (RBV) increased cytotoxicity in HepG2 cells [6]. We recently investigated mitochondrial genotoxicity among people who inject drugs (PWID) who were treated with a combination of SOF, DCV, and RBV drugs. This population is well known for its characteristically high prevalence of HCV infection and constitutes one of the key target populations to achieve HCV elimination, as stipulated by the World Health Assembly in the first Global Health Sector Strategy on Viral Hepatitis 2016–2021 [7]. Moreover, this population is continuously exposed to several other genotoxicants that might increase the risk of genetic instability [8]. Overall, we found that the variation in mtDNA copy per cell (MCN) did not differ significantly before and after treatment, while the proportion of detectable mtDNA deletions (MDD) increased after treatment. Combined exposition of DAA with other illicit drugs may explain some of these variations [8].

If mtDNA copy number can fluctuate with time, elimination of deleted DNA molecules or defective mitochondria is a complex process and could instead tend towards an accumulation over time, leading to bioenergetic defects, mito-ageing processes, and, possibly, to cell transformation [9]. Herein, we conducted an observational study by measuring the mitochondrial parameters nine months after the end of treatment among HCV-cured PWID. We then tested whether these parameters were similar to those of an HCV-seronegative PWID population.

## 2. Materials and Methods

### 2.1. Study Design and Study Population

The design of the study is a prospective cohort study. The study population was made up of HCV RNA-positive PWID who were successfully treated with combinations of sofosbuvir, daclatasvir, and/or ribavirin for 12 weeks (negative sustained viral response at 12 weeks, known as the SVR12 HCV viral load) from the DRIVE-C study (NCT03537196) [10]. An HCV-seronegative PWID control population was obtained from the initial HCV screening phase of the previously mentioned study. PWID were recruited during a respondent-driven sampling survey (RDSS) conducted by community workers in Hai Phong, Vietnam, in 2018. This RDSS enrolled a total of 1444 PWID [10].

### 2.2. MtDNA Genotoxicity Assays

A detailed description of the assays has previously been published [8]. Succinctly, blood samples were collected on dried blood spot cards (DBS, WhatmanTM 903, GE Healthcare Bio-Sciences Corp.) and DNA was extracted using QIAamp DNA Mini kit (Qiagen, Courtaboeuf, France). Mitochondrial copy number (MCN) was assessed by real-time quantitative PCR using QuickScanTM Mitox kit (Primagen©, Amsterdam, The Netherlands). The MCN is expressed as the number of mtDNA cp/cell. The percentage of MDD was obtained by relative quantification of two qPCRs (2^-ΔΔCt^ method), one targeting the region which encompasses more than 85% of the known mtDNA deletions and one targeting a very constant region, using DNA extracted from plasma-rich platelets as a calibrator. The MDD rate is the ratio of mutated mtDNA to total mtDNA and is expressed as a percentage [8].

### 2.3. Statistical Analysis

Baseline characteristics are described as frequencies with percentages for categorical variables, or as means with their standard deviation for continuous variables. MtDNA parameters are expressed as raw values for MCN or as percentages of MDD, with their 95% confidence interval. Median MCN was calculated for each of the three time points and subsequently compared between time points, as well as median MCN of PWID with an initial HCV negative serology (n = 260). The threshold for statistical significance was set for a *p* value < 0.05.

For the risk factor analyses, variations in MCN and MDD for each PWID were calculated between the end of treatment and 36-weeks post treatment discontinuation. PWID were next stratified into two classes depending on their ΔMCN and their ΔMDD results, using a threshold set at the first tercile value of the pooled data, equivalent to ≤ −76.5% for ΔMCN and to ≤ −32% for ΔMDD. To construct multivariate models, we selected variables with a *p* ≤ 0.20 in univariate analyses among demographic, drug consumption use, co-infections, and co-medication data. The final model was constructed in a stepwise manner and validated by considering the smallest AIC (Akaike information criteria). The threshold for statistical significance was set for a *p* value < 0.05. Statistical analyses were performed on SAS^®^ studio (Copyright © 2012-2020, SAS Institute Inc., Cary, NC, USA).

### 2.4. Ethics Approvals

Participants signed an informed consent form at enrolment that included the use of their samples in ancillary studies related to HCV infection among PWID. The present study complies with the Declaration of Helsinki and Good Clinical Practice, was approved by the Scientific Advisory Board of DRIVE-C, and subsequently by the Institutional Review Board of the Haiphong University of Medicine and Pharmacy, Vietnam (#01/HPUMPRB).

## 3. Results

### 3.1. Study Population

Out of the 332 PWID with mitochondrial data at the end of treatment, 297 attended the 9-month follow-up visit and 295 had a full set of paired data (Appendix A). Reasons for not attending the 9-month follow-up visit were primarily “being incarcerated”. HCV-treated PWID included in these analyses (n = 295) differed from those not included (n = 37), with less frequent HIV-positive infection statuses and being less frequently engaged in a methadone program (data not shown). PWID were almost exclusively men (97.6%), with a mean age of 42.0 years, and were administered SOF400/DCV60 (n = 149), SOF400/DCV90 (n = 119), or SOF400/DCV/RBV (n = 27) (Table 1). At the end of the study, 121 (41.0%) PWID reported still injecting heroin, 243 (82.4%) being under methadone therapy, and 33 (11.2%) smoking methamphetamine. All HIV-infected PWID had received ARV treatment, and none had previously been treated for HCV infection. The HCV-seronegative PWID group was characterized by less HIV infection (2.3% vs. 46.4%), a more recent history of injection (injecting for less than 5 years: 22.3 % versus 4.4%), and a greater consumption of methamphetamine (72.7% vs. 64.6%) (Table 1).

### 3.2. Long-Term Dynamics of the mtDNA Parameters

Nine months after the end of treatment, median MCN dropped from 568.7 copies/cell (95%CI: 494.5; 647.7) to 184.0 (95%CI: 168.6; 198.9), and median MDD increased from 0.35 (95%CI: 0.32; 0.39) to 0.49 (95%CI: 0.45; 0.52). These values were statistically different from both the baseline values of HCV-infected PWID and the baseline values of control HCV-seronegative PWID (Table 2).

### 3.3. Determinants of Altered mtDNA Parameters

People who inject drugs were next stratified by those being in the highest tercile of MCN loss (with a threshold set at 76.5% loss or more) or those in the highest tercile of increase in MDD rate (with a threshold set a 32% loss or more) versus all those remaining. Multivariate analysis showed that none of the current exposure factors including medications, drug consumption, or viral co-infections are associated with an increased risk of being “PWID with high loss of MCN” or being “PWID with a high accumulation of MDD”, in the 9-month period following the end of treatment. Furthermore, having been exposed to one DAA regimen compared to another was not associated with mitochondrial genomic instability (Appendix A).

## 4. Discussion

A 9-month follow-up of DAA-treated PWID revealed that mitochondrial parameters are not yet stable; the percentage of mtDNA molecules with deletion increased, while the number of copies per cell dropped drastically. We recently reported an increased rate of MDD upon completion of DAA treatment, while remaining subclinical. The present data supported the long-term persistence of MDD, which even worsened, while once again remained below the clinical level. Even among infected PWID that were treated and cured from the HCV infection, nine months after DAA treatment discontinuation these parameters did not revert back to levels observed among uninfected individuals.

MtDNA mutations and deletions lead to mitochondrial dysfunctions. These processes participate in biological ageing and age-related diseases (such as neurodegenerative and cardiovascular pathologies and cancers [11,12]). They have already been well illustrated among people living with HIV (PLHIV) undergoing antiretroviral therapy. In fact, PLHIV have a shorter life expectancy than the general population and are more prone to age-related diseases [9]. Cellular processes exist to eliminate defective mitochondria [11,12]. Beyond a threshold which is not yet defined, and which may vary from one cell type to another, and most likely with age, the cells undergo cell death. The direct role of ARV in this toxicity has recently been demonstrated. Persons initiating post-exposure prophylaxis after a non-occupational sexual exposure to HIV showed mitochondrial toxicity, which was worsened for those having received one-month regimens containing zidovudine (AZT) molecules [13]. The clearance rate for these alterations has not yet been defined, but initial arguments suggest that it may be very slow. Exposure to ARV drugs during pregnancy was assessed in terms of mitochondrial toxicity in *Patas* monkey pups at specific time points from birth. Noticeably, mitochondrial toxicity was still detectable at three years of age, which corresponds to approximately 15 years of age in humans [14,15]. The same observation was conducted for the number of mtDNA copies in HIV-uninfected infants born to HIV-positive mothers undergoing ARV treatment. Aldrovrandi et al. showed that the mtDNA copy number was only able to return to baseline values at five years of age [16]. Given that DAAs are of the same class of polymerase inhibitors as ARVs, the mitochondrial genotoxicity reported herein could follow these same mechanisms.

It is noteworthy that slow cycling cells are known to accumulate more deleted mtDNA molecules than rapid cycling cells, such as blood cells, suggesting that the MDD rate in other cell types in DAA-treated patients may be even higher [11,17,18]. Given that we were not able to identify other exposures as risk factors for the observed mtDNA instability during the follow-up period, DAA treatments may have primed the acquisition of MDD, which then persists and continues to replicate.

MtDNA instability has been previously associated with HCC, but its role in the pathophysiology has not yet been established [19]. Reported side effects of DAA treatments are minimal so far, but HCC recurrence is currently under scrutiny. Further investigation is required regarding the level of MDD accumulation after DAA treatment compatible with HCC development.

This study has several limitations. First, it is a monocentric study. Secondly, the before–after study design allowed us to report observational data only. Analyses were conducted at the participant level, so that each participant was its own control. The risk factor analyses cannot be used to decipher the mechanism underlying the mitochondrial genotoxicity but rather addressed putative concomitant exposures. Thirdly, the PWID population analysed herein is almost exclusively men. Given the number of women, we were not able to conduct a stratified analysis to observe these effects specifically in women. In addition, our conclusions should only be applied to PWID. To increase the scope of our findings and be able to observe gender-specific effects, the present study deserves to be reproduced among non-drug users of both sexes. Fourthly, the integrity of the mtDNA molecules was only investigated through the search of their common deletion and not in terms of point mutation. Other regions of the mtDNA may also be targeted for deletions, given that the drug exposures were particular. The latter two aspects would require sequencing approaches.

Altogether, these findings strongly suggest the persistence of mitochondrial dysbiogenesis after HCV treatments among male PWID, which remained subclinical over the course of the 9-month follow-up. The presented data call for a longer follow-up of this mtDNA instability.

## Figures and Tables

**Table 1 biomedicines-10-02541-t001:** Baseline characteristics of PWID included in the analysis and compared to HCV-seronegative PWID.

	HCV-Cured PWID N= 295	HCV-Seronegative PWID N= 260	*p*-Values
**DEMOGRAPHIC DATA**			
**Sex**, Male or transgender, n (%)	288 (97.6)	246 (94.6)	0.06 *
**Age**, years, mean (SD)	42.0 (7.4)	41.7 (10.1)	0.74 #
**VIRAL INFECTIONS**, n (%)			
HIV coinfection	137 (46.4)	6 (2.3)	<0.001 *
HBV coinfection	18 (6.1)	N.A.	-
**TREATMENTS DAA**, n (%)			
SOF400/DCV60	149 (50.5)	N.A.	
SOF400/DCV90	119 (40.3)	N.A.	
SOF400/DCV/RBV	27 (9.1)	N.A.	-
**ARV**			
Receiving ARV treatment	137 (46.4)	3 (1.2)	< 0.001 *
**SUBSTANCE USE—Heroin**, n (%)			
Number of years of injection			
Less than 5 years	13 (4.4)	58 (22.3)	
5 to 10 years	57 (19.3)	80 (30.8)	
10 to 15 years	78 (26.4)	63 (24.2)	
Over 15 years	147 (49.8)	59 (22.7)	<0.001 *
Frequency of injection per month			
Less than once a day	88 (29.8)	90 (34.6)	
Daily	207 (70.2)	170 (65.4)	0.23 *
**Methamphetamine**			
Urinary test positive at baseline	79 (26.8)	93 (35.8)	0.02 *
Declaration of consumption	190 (64.4)	189 (72.7)	0.04 *
Frequency of consumption per month			
<4 times per month	142 (48.1)	121 (46.5)	
≥4 times per month	48 (16.3)	68 (26.1)	0.008 *
**Tobacco smoking**	286 (96.9)	N.A.	-
**Hazardous drinking ^£^**	75 (25.4)	84 (32.3)	0.07 *

*: Chi-squared test; #: *t*-test; £: score above 4 for men or 3 for women on the AUDIT-C scale; N.A.: not available.

**Table 2 biomedicines-10-02541-t002:** MtDNA parameters among HCV-treated PWID and control PWID.

Mitochondrial Outcomes	HCV-Treated PWID (n = 295)			Control PWID (n = 260)
	Baseline	End of Treatment	9-Month Follow-Up	
MCN (c/cell)	481.2 (448.6; 524.6)	568.7 (494.5; 647.7)	184.0 (168.6; 198.9)	439.1 (405.9; 466.8)
MDD	0.26 (0.23; 0.29)	0.35 (0.32; 0.39)	0.49 (0.45; 0.52)	0.31 (0.24; 0.34)

Values are medians with 95% confidence interval; MCN: mitochondrial DNA copy number; MDD: mitochondrial DNA deletion rate.

## Data Availability

The data presented in this study are available on request from the corresponding author.

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
