# Peer review of "Long-Term Persistence of Mitochondrial DNA Instability among HCV-Cured People Who Inject Drugs"

_biomedicines, 2022, doi:10.3390/biomedicines10102541_

Round 1

Reviewer 1 Report

Accept in present form

Author Response

Accept in present form

Thank you for supporting the manuscript

Reviewer 2 Report

The Authors found that HCV treatments among males that inject drugs may induce mitochondrial dysbiogenesis. This conclusion has been obtained by looking at the mtDNA copy per cell (MCN), and mtDNA deletions (MDD) in DNA extracted from blood.

The following points should be addressed:

1)      Please state abbreviations the first time they appear, even in the abstract (i.e. direct-acting antivirals 41 (DAA)).

2)      Please indicate the statistic test used for univariate analysis (parametric/non-parametric).

3)      Please explain why a P 0.2 has been chosen as a threshold in univariate analyses

4)      Please indicate the P-value threshold considered statistically different and the test used to obtain the P-values.

Author Response

The Authors found that HCV treatments among males that inject drugs may induce mitochondrial dysbiogenesis. This conclusion has been obtained by looking at the mtDNA copy per cell (MCN), and mtDNA deletions (MDD) in DNA extracted from blood.

The following points should be addressed:

  • Please state abbreviations the first time they appear, even in the abstract (i.e. direct-acting antivirals 41 (DAA)).

The text has been checked through out to answer the comment.

  • Please indicate the statistic test used for univariate analysis (parametric/non-parametric).

The statistic tests are now inserted in the Tables.

  • Please explain why a P ≤ 0.2 has been chosen as a threshold in univariate analyses

We used this threshold for variable selection for the multivariate analysis in order to limit the adjustments. It was not used as a threshold for significance. We modified the sentence to avoid confusion as follows: “To construct multivariate models, we selected variables with a p ≤ 0.20 in univariate analyses among demographic, drug consumption use, co-infections and co-medication data.”

  • Please indicate the P-value threshold considered statistically different and the test used to obtain the P-values.

This information is now added to the text, Line 98-99 & 108-109.

Reviewer 3 Report

This is a study on the long-term effects of DAA therapy on HCV-infected drug users. It is however an observational study, with no information on the underlying mechanisms of the effect of DAAs on mitochondrial DNA instability.  Also, I think the findings are preliminary and the authors should address the inherent lack of lines of evidence, they identify as disadvantages of the study, in order to provide the reader with clearer conclusions.

Finally, it would be interesting to know if the patient groups used for the  study comprised treatment naive patients or have been treated before with intereferon-based regimes. If that is the case, this should be included as a prediction variable in their statistical models.

Author Response

This is a study on the long-term effects of DAA therapy on HCV-infected drug users. It is however an observational study, with no information on the underlying mechanisms of the effect of DAAs on mitochondrial DNA instability.  Also, I think the findings are preliminary and the authors should address the inherent lack of lines of evidence, they identify as disadvantages of the study, in order to provide the reader with clearer conclusions.

We fully agree with the reviewer that the manuscript is an observational study completed with a risk factor analysis to identify why some have worse prognoses compared to others. We modified the text to more clearly describe the study design. Deciphering mechanisms was not the objective of the study and we thought that it would have been too speculative to discuss it. We revised the last § of the introduction as follows: Line 68-69 “Herein, we conducted an observational study by measuring the mitochondrial parameters nine months after the end of treatment among HCV-cured PWID” and the limitation paragraph as follows: Line 199-204 “Secondly, the before-after study design allowed us to report observational data only. Analyses were conducted at the participant level, so that each participant was its own control. The risk factor analyses cannot be used to decipher the mechanism underlying the mitochondrial genotoxicity but addressed putative concomitant exposures.

Finally, it would be interesting to know if the patient groups used for the study comprised treatment naive patients or have been treated before with intereferon-based regimes. If that is the case, this should be included as a prediction variable in their statistical models.

This information is now added to the text, Line 127-128 “All HIV-infected PWID received ARV treatment and none of them were previously treated for HCV infection.” In Vietnam, IFN-based regimen was very expensive and not affordable to PWID (not paid by the health insurance). As a matter of fact, no HCV diagnostic screening among this population was conducted until we implemented the research program. Consequently, none of them had previously received any HCV treatment.

English language and style was edited by a US native speaker.

Round 2

Reviewer 3 Report

I am happy with the revised version.